# Habitat Quality Assessment and Ecological Risks Prediction: An Analysis in the Beijing-Hangzhou Grand Canal (Suzhou Section)

Yixin Zhang [1,2,3,]*, Chenyue Zhang [1], Xindi Zhang [1,2], Xinge Wang [1], Ting Liu [1,2], Zhe Li [1,2], Qiaoyan Lin [4], Zehui Jing [1,2], Xinyue Wang [1], Qiuyue Huang [1,2], Wenxin Sun [1], Jun Zhai [1], Li Tan [1,2], Jieqing Wang [1,2], Guoyan Zhou [1], Yasi Tian [1], Jianli Hao [4], Yu Song [4] and Fei Ma [4]

1 Department of Landscape Architecture, Gold Mantis School of Architecture, Soochow University, Suzhou 215123, China
2 China-Portugal Belt and Road Cooperation Laboratory of Cultural Heritage Conservation Science, Research Center of Landscape Protection and Ecological Restoration, Soochow University, Suzhou 215006, China
3 Zhejiang Institute of Research and Innovation, The University of Hong Kong, Hong Kong, China
4 Department of Environmental Sciences, Xi'an Jiaotong-Liverpool University, Suzhou 215123, China
* Correspondence: yixin.zhang2019@suda.edu.cn

**Abstract:** With the fast pace of global urbanization, anthropogenic disturbances not only lead to frequent disasters, but also cause direct and indirect ecological and economic losses. To reduce the adverse effects of anthropogenic disturbances as part of sustainable ecosystem management, assessments of habitat quality and ecological risk are necessary. The objectives of this study are to analyze environmental conditions of the Beijing-Hangzhou Grand Canal (Suzhou section) for evaluating habitat quality and habitat degradation, and to conduct ecological-risk early warning assessment in this section. The Grand Canal is the longest and first canal in the world to be artificially excavated from natural rivers and lakes. By evaluating habitat quality using the InVEST suite of open-source software models for mapping and valuing the ecosystem, it was found that the natural lands with high habitat quality such as wetlands, forests and lakes along the Suzhou section of the Grand Canal have gradually decreased, while construction lands such as roads and buildings have gradually increased; there is a clear trend of decreasing areas with high habitat quality and increasing areas with low habitat quality, which is likely the result of urbanization. It was also found that the region has a high habitat degradation index, meaning that areas located at the junction of different land types are vulnerable to the surrounding environment due to narrow buffer zones that allow areas with high habitat quality to be easily affected by areas with low habitat quality. In terms of ecological risks, it was found that the natural land area with high habitat quality in the downstream locations was declining, thereby increasing the risks of pollution and flooding events while reducing the ecosystem's resilience. The valuation model used in this study can be used as an effective decision-support tool to prioritize important ecological areas for conservation in the Grand Canal, and can also be adapted for use in the ecosystem management of other regions.

**Keywords:** InVEST model; habitat degradation; urbanization; ecological risks; ecosystem management

## 1. Introduction

A healthy environment is the foundation of biodiversity and of human survival [1]. The materials and services produced through ecosystem processes have irreplaceable value for human beings with direct and indirect benefits [2]. Running water ecosystems, as one of the most diverse and dynamic ecosystems, provide rich goods and services for humans, such as land conservation, soil purification, creating regional microclimates, providing habitat, and increasing biodiversity [3]. However, with the acceleration of urbanization that has significantly driven climate change and pollution, running water ecosystems have been

seriously damaged, threatening habitat quality and biodiversity and increasing ecological risk. Habitat quality assessment and the environmental mechanism involved in the process of ecological risk provide the practical basis and theoretical guidance for the prediction of ecological risk.

Habitat quality is the availability of living resources in a specific environment and the ability of the environment to provide suitable living conditions for individuals or populations [4]. It is regarded as a key representation of regional biodiversity and ecosystems, and a key link to ensure regional ecological security and benefit human well-being [5,6]. Studying the temporal and geographical aspects of habitat quality in regional rivers, as well as comprehending their dynamic regulations, is not only the foundation for developing river-based sustainable development plans, but also an essential prerequisite for land-use planning and management [7].

The methods of habitat quality assessment usually vary according to the scale involved. For large-scale habitats, functional models, such as regression model [8], niche model [9], and InVEST model [10], are used to evaluate the habitat quality by representing the overall structure with the parameters related to the habitat's physical and chemical characteristics. For small-scale habitats, the index evaluation based on field investigation is usually used: (1) bio-habitat suitability model based on the correlation between aquatic organisms (mainly benthic animals and plants) and river habitats, such as RIVPACS (River Invertebrate Prediction and Classification System) in the UK [11]; and (2) model based on the physical structure of river characteristics, such as AUSRIVAS (Australian River Assessment System) [12]. With more attention being paid to the environment and the prominent benefits of high-quality river habitats, how to evaluate the quality of a river habitat in order to solve practical problems is an important topic in the field of ecological risk prediction and ecological restoration [13].

Ecological risk refers to the possibility of structural and functional damage to the ecosystem, which will reduce the health, productivity, genetic structure, economic value, and aesthetic value of the ecosystem at present and in the future [14]. According to its occurrence characteristics, it can be divided into sudden risks (such as flooding and the leakage of flammable materials) [15] and cumulative risks (such as heavy metal pollution) [16] (see Figure 1). The possibility of ecological risk is often related to habitat quality because ecosystem services closely related to habitat quality can improve ecosystem resilience and reduce the risk of disturbance.Specifically, the system is generally in a stable state under slight human disturbance due to the resilience of the habitat system itself [17]. With the increase inhuman disturbance intensity, the habitat quality gradually decreased, and the carrying capacity of the system to resist ecological risks gradually decreased [18]. Therefore, faced with potential ecological risks, especially cumulative risks in the future, it is necessary to evaluate the current habitat quality and predict the future one according to the temporal and spatial characteristics of land use. It is then possible to predict ecological risks in a comparative way based on the results.

The Beijing-Hangzhou Grand Canal is one of the most outstanding water conservancy projects in the history of human civilization, which has had a profound impact on the economic development and social and cultural civilization in this region. The Grand Canal is a vast inland waterway system in China, running from Beijing in the north to Hangzhou in the south, which was constructed in sections from the 5th century BC onwards, and created as the world's most extensive and largest civil-engineering project prior to the Industrial Revolution. At the 38th World Heritage Congress in 2014, the Grand Canal was added to the World Cultural Heritage List. With the development of the city, the Beijing-Hangzhou Grand Canal is faced with the deterioration of water environment, and the safety of the river ecological environment is facing severe challenges. In recent years, various environmental emergencies have occurred, including the ecological hidden dangers brought about by urbanization.

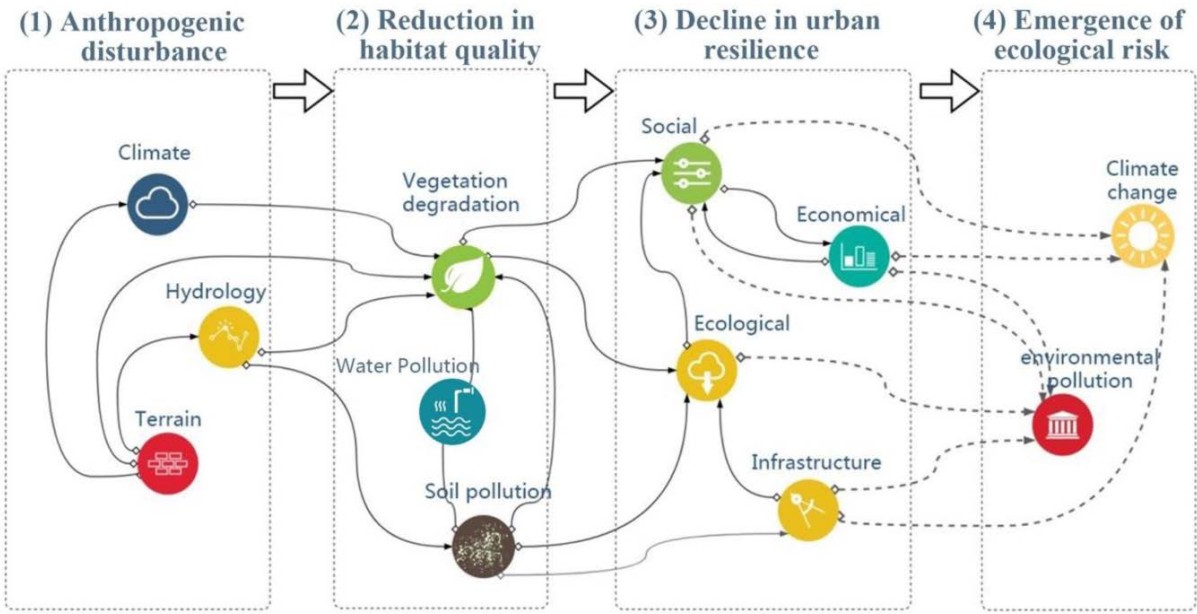

**Figure 1.** Anthropogenic disturbance (1) induced decline in habitat quality (2) leading to a reduction in ecosystem resilience (3) and an increase in ecological risks (4).

Although there are many studies on habitat quality assessment, there is a lack of research on the ecology of protected canals with a history of nearly 2500 years, and the ecological risk area has not been considered. Ecological risk prevention and habitat quality improvement of rivers with a long history are important contents of cultural protection and ecological environment protection in the area.

This study investigated the Suzhou section of the Beijing-Hangzhou Grand Canal for assessing habitat quality by using the InVEST model to analyze remote sensing maps with habitat features. The objectives of this study are to (1) evaluate the present habitat quality and forecast the future of the canal and (2) predict the possible ecological risks, so that efficient ecosystem management can be conducted in the future.

## 2. Conceptual Framework of the Relationship between Habitat Quality and Ecological Risks

Ecosystem habitat quality refers to the ability of a species' living space to provide suitable conditions for individuals or populations to survive [19]. Evidence suggests that ecosystems have a certain ability to recover themselves, which is related to habitat quality [20]. Based on this, ecosystems can gradually return to a stable state after the influence of sudden ecological disturbances or cumulative ecological disturbances, which is ecosystem resilience [20–22].

However, with rapid urbanization, more and more natural land has been transformed into construction land [23,24], which has caused habitat fragmentation and biodiversity loss. At the same time, habitat quality has also declined, which has brought about an increase in ecological risk impacts, and it makes ecological problems more frequent (see Figure 2).

These ecological problems mainly refer to factors such as pollutant leakage and illegal discharge, natural disasters, production safety accidents, etc., which cause pollutants and other toxic and harmful substances to enter the atmosphere, water, soil, and other environmental media in a short period of time, causing sudden or potential environmental damage. Risks (including production, storage, transportation and use of chemicals, environmental risks due to illegal activities, eutrophication, heavy metal pollution in water and soil, secondary sudden ecological risks caused by natural disasters brought about by climate change, etc.) endanger public health and property safety, cause ecological environment damage, and have major negative social impacts.

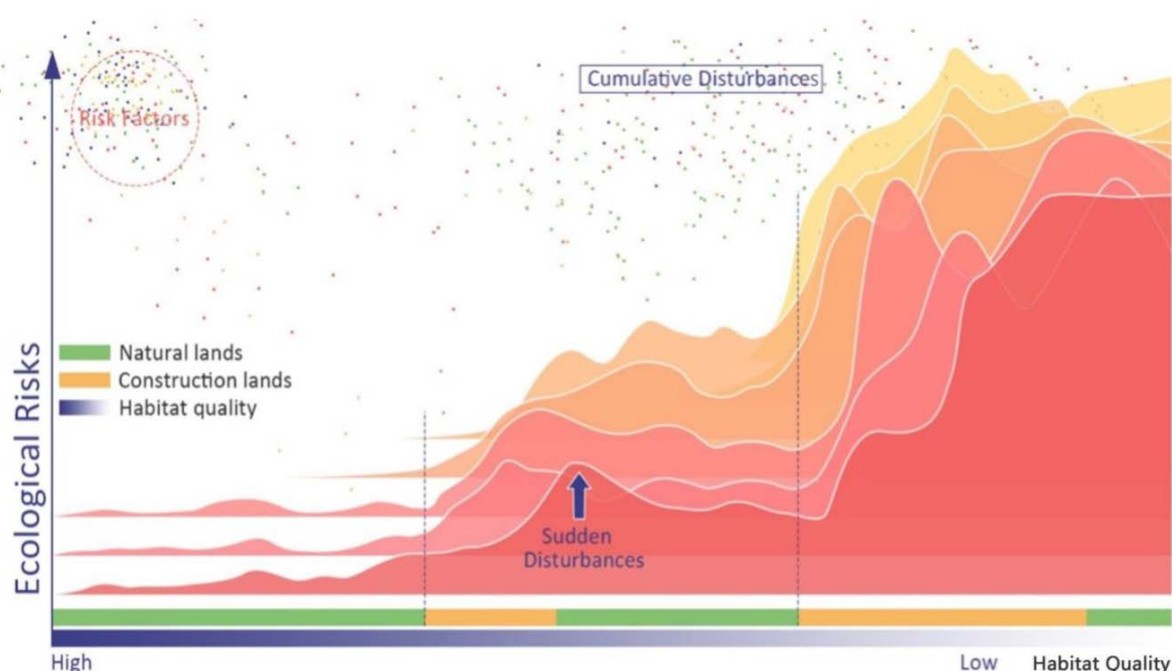

**Figure 2.** Conceptual framework for the relationship between habitat quality and ecological risks.

There is a certain negative correlation between habitat quality and ecological risks. As construction land increases for such things as roads, factories, houses, and so on, the habitat quality will decrease. At the same time, ecosystems will face greater and more frequent ecological risks, and the harm caused by anthropogenic disturbance will become more significant. According to the characteristics of ecological risk occurrence, they can be divided into an abrupt change of disturbances (sudden disturbance) and cumulative disturbances. Cumulative disturbances will bring more lasting and higher potential ecological risks, while sudden disturbances often produce great harm in a short time.

## 3. Materials and Methods

### 3.1. Study Area

The study area is the buffer zone within 400 m of the Beijing-Hangzhou Grand Canal in the Suzhou Municipal District,120°44′E—121°00′E, 31°26′N—31°45′ N. The total length of the Beijing-Hangzhou Grand Canal in Suzhou is about 95 km (the old canal in the south of Taipu River is about 13km long), accounting for 40% of the canal in southern Jiangsu. The study area is flat, low-lying, and 3–5m above sea level; it has a subtropical monsoon oceanic climate, with an annual average temperature of 16–18°C and an annual precipitation of about 1000–1400 mm.

The Grand Canal witnessed the growth and development of Suzhou and promoted its prosperity. The Suzhou section of the Beijing-Hangzhou Grand Canal spans five administrative districts in Suzhou with a total area of about 1561km². Various types of land are relatively mixed, mainly for residential, industrial, and storage. Suzhou's "one park and three districts" (Suzhou Industrial Park, Suzhou High-tech Zone, Wuzhong Development Zone, and Wujiang Development Zone) all rely on the canal. The development of the canal provides unique and advantageous conditions for the rapid rise of various development zones in Suzhou (see Figure 3).

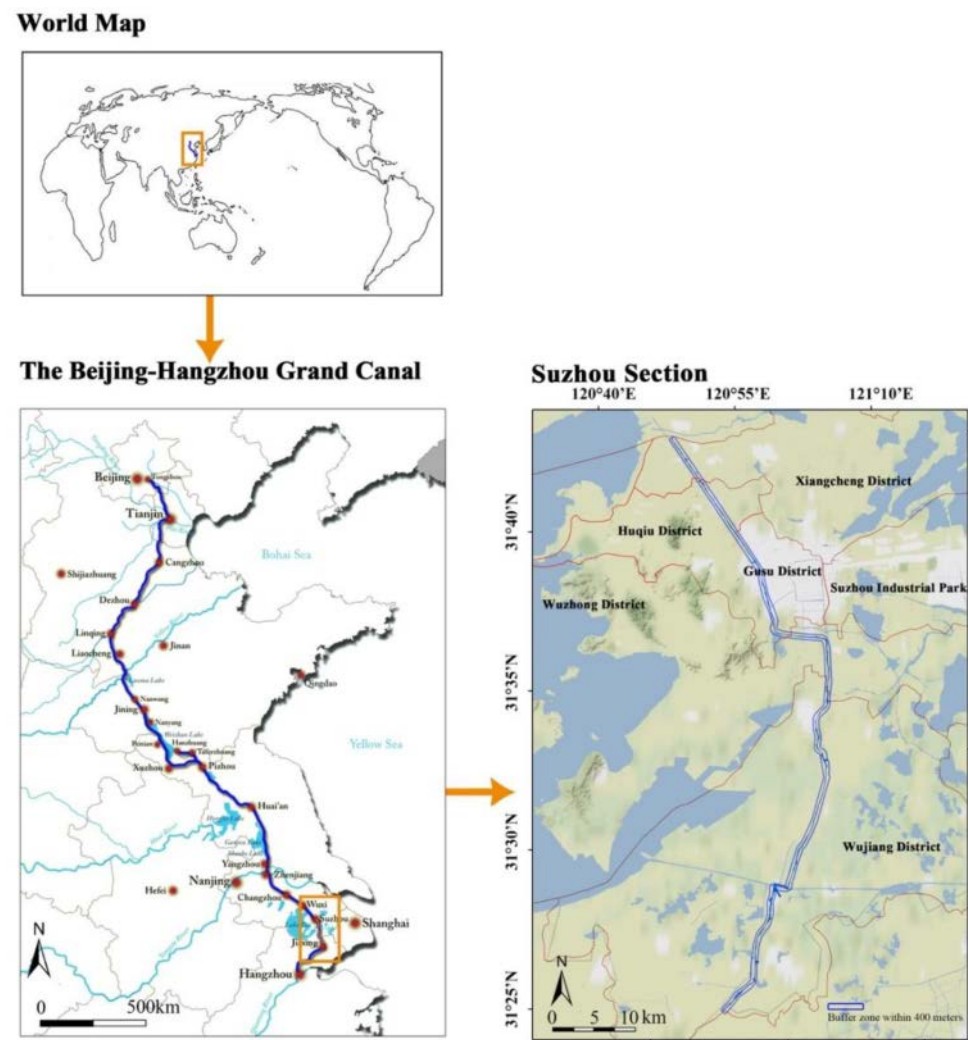

**Figure 3.** Location of study area (source: https://wuu.wikipedia.org/wiki/, accessed on 16 December 2021).

*3.2. Data Source and Preparation*

In this study, Landsat 8 remote sensing data of Suzhou in 2010 and 2018 were obtained from open data source LandsatLook (https://landsatlook.usgs.gov/explore. accessed on 18 December 2021). Mosaic registration correction and unsupervised classification with Envi was used for the present situation of LULC (the land use/land cover) in Suzhou city, with a spatial resolution of 30 m. Guidos Toolbox is used to convert the land use map into TIFF data for analysis. The spatial resolution of digital elevation model (DEM) data is 30m, which comes from geospatial data cloud (http://www.gscloud.cn/, accessed on 18 December 2021). According to the classification of LULC status (GB/T 21010-2017) and research purposes, the landscape types in the study area are divided into cultivated land (paddy fields and cropland), woodland (woodland, bush, open woodland, and other woodland), grassland (high-coverage grassland), wetland (rivers, lakes, shallows, beaches, and swamps), construction land (urban land rural residential area and industrial mining and transportation land), and bare land (bare land and barren land). Within a 400 m buffer zone of the Suzhou section of the Beijing-Hangzhou Grand Canal, there are eight land use types—lakes, rivers, paddy field, shallows, industrial mining and transportation land, urban land, rural residential area, and other woodland. Lakes refers to the land below the perennial water level in the naturally formed water accumulation area. Rivers refers to the land below the perennial water level of natural or artificially excavated rivers. Paddy field refers to the cultivated land with water source guarantee and irrigation facilities, which

can be irrigated normally in ordinary years, and used for planting aquatic crops such as rice, including the cultivated land where rice and dry land crops are planted in rotation. Shallows refers to the water surface surrounded by the shoreline of the normal water storage level of the reservoir and the water surface of the shallows with a water storage capacity of less than 100,000 m$^3$. Other woodland including open woodland, immature woodland, cut-over land, and nursery (see Figure 4).

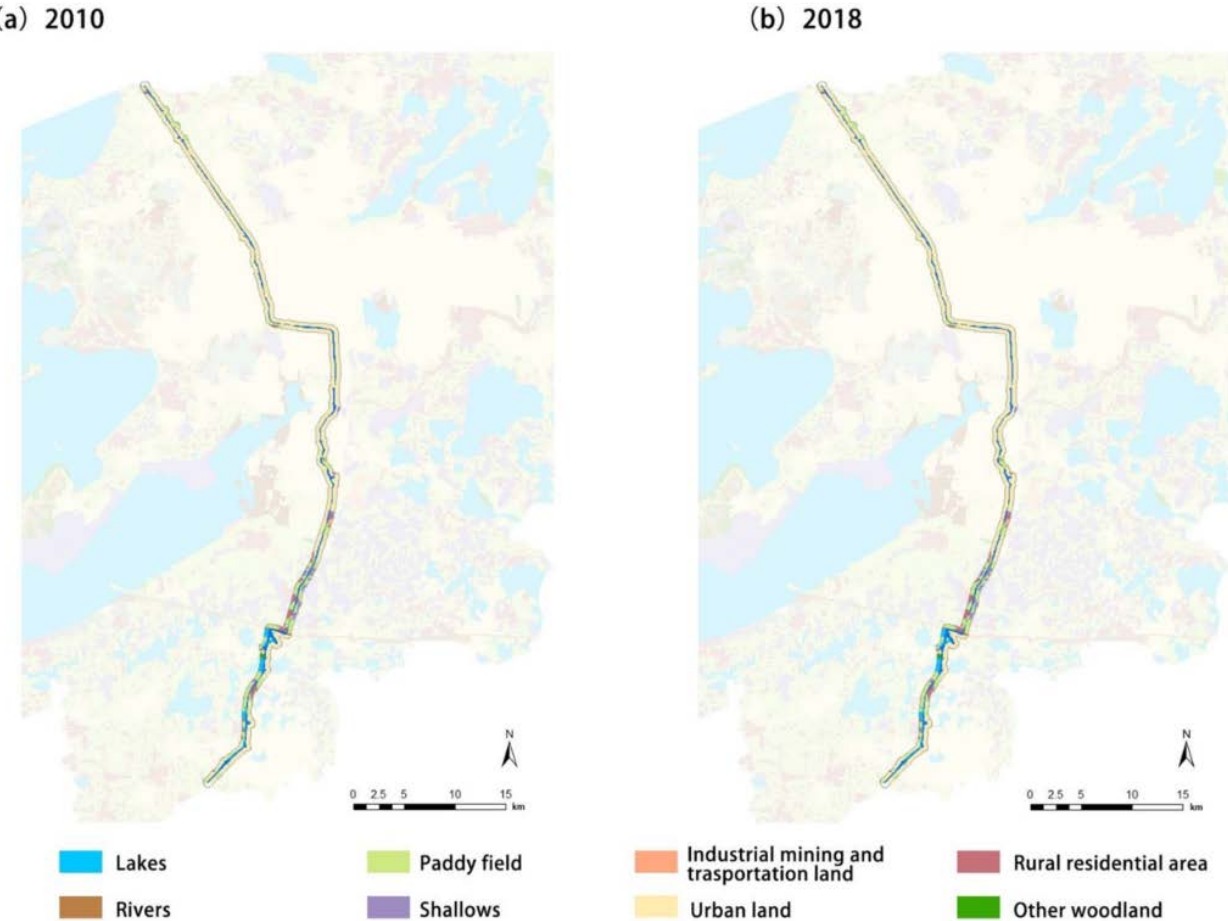

**Figure 4.** Grand Canal land use status in in 2010 and 2018 (eight land use types with urban land the main type of land use).

*3.3. Description of Habitat Quality Assessment by InVEST Model*

3.3.1. Description of the Habitat Quality Model

This study uses the habitat quality model of InVEST (3.9.2) to evaluate the habitat quality of the Beijing-Hangzhou Grand Canal (Suzhou section). Habitat quality is spatially heterogeneous, so it could combine information on LULC and threats to biodiversity to produce habitat quality maps and habitat degradation maps [25]. The InVEST habitat quality (InVEST HQ) module has been widely used to calculate metric with several conservation planning purposes. The InVEST HQ module has been used to assess historic and dynamic changes in habitat quality [10,26] in Italy [27], China [28], and Nicaragua [29].

Habitat quality refers to the ability of the ecosystem to provide conditions appropriate for individual and population persistence, and is considered a continuous variable in the model, ranging from low to medium to high [25]. Habitat quality depends on a habitat's proximity to human land uses and the intensity of these land uses [25]. Modeling InVEST HQ module is mediated by four factors: (1) the relative impact of each threat, and the importance of one threat compared to the others [25]; (2) the maximum distance between the land cover and the threat source beyond which the threat does not affect habitat quality [30]; (3) the habitat suitability, the suitability of a land cover class to provide habitat

for living things [31]; (4) the relative sensitivity of each habitat type to each threat on the landscape [25]. Figure 5 illustrates how it works.

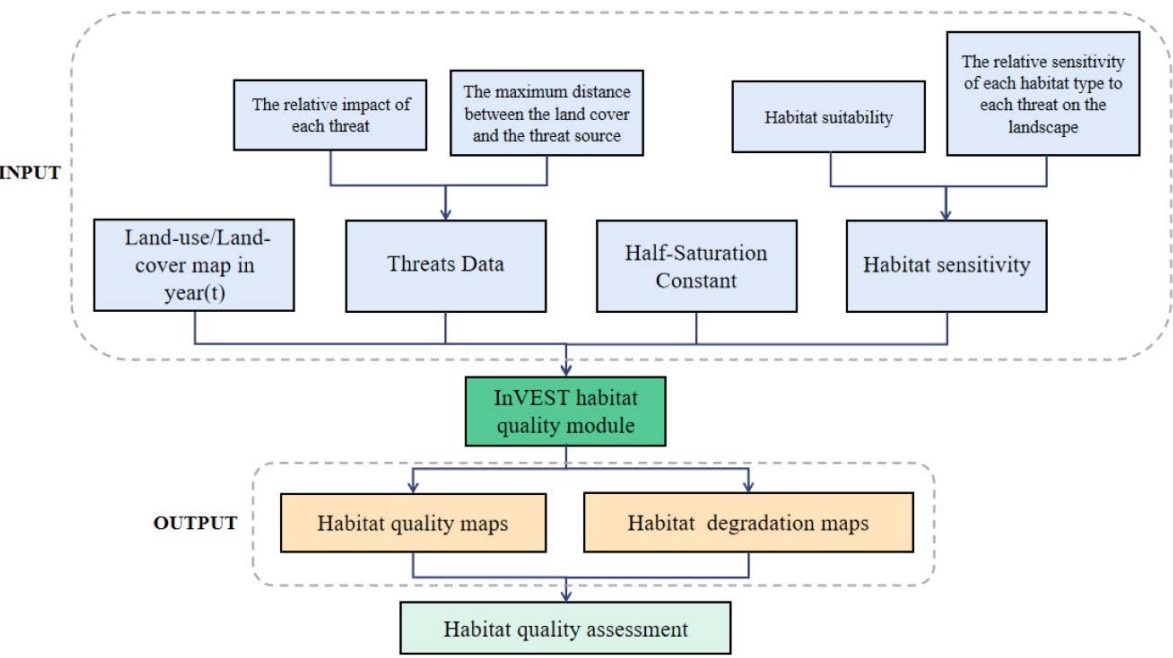

**Figure 5.** Working framework of InVEST habitat quality module, t stands for the study year.

The InVEST HQ module evaluates habitat quality as a function of threats using a half saturation curve to convert habitat degradation to habitat quality score [25]. Habitat quality assessment includes the calculation of habitat quality index and habitat degradation index. The specific equation is as follows:

1.  Habitat Degradation Index

The value of the habitat degradation index represents the impact level of threat factors on habitat and, thus, the potential for habitat destruction and the degradation of habitat quality. The habitat degradation index (Equation (1)) takes values from 0 to 1; the closer the value is to 1, the greater the potential damage caused by the threat source to the regional habitat and the more detrimental the maintenance of biodiversity [25,32].

$$D_{xj} = \sum_{r=1}^{R} \sum_{y=1}^{Y_r} \left( \frac{W_r}{\sum_{r=1}^{R} W_r} \right) r_y i_{rxy} \beta_x S_{jr}. \quad \right) \tag{1}$$

Shown as Equation (1): $r$ refers to stress factor; $R$ refers to the number of stress factors; $Y_r$ refers to the grid number of stress factors; $W_r$ refers to the weight of stress factors; $r_y$ refers to the number of stress factors corresponding to a single grid cell; $\beta_x$ refers to the accessibility level of grid cell $x$, ranging from 0 to 1; $i_{rxy}$ refers to the influence distance of the stress factor value $r_y$ of grid unit $y$ on grid unit $x$, which can be divided into linear decay and exponential decay [25].

2.  Habitat Quality Index

The habitat quality index (Equation (2)) changes continuously from 0 to 1. The closer the value is to 1, the more favorable the maintenance of biodiversity [25,32].

$$Q_{xj} = H_j \left( 1 - \left( \frac{D_{xj}^z}{D_{xj}^z + K^z} \right) \right) \tag{2}$$

where $Q_{xj}$ refers to the habitat quality index of grid unit x in land use type j; $H_j$ refers to the habitat suitability of land use type j, ranging from 0 to 1; z is the scale constant; k refers to half-saturation constant, which is generally set to 0.5. It should be customized according to the data; $D_{xj}$ refers to the habitat degradation index of grid unit x in land use type j [25].

### 3.3.2. Parameter Requirement Analysis of the InVEST Model

Through the working framework of the InVEST habitat quality module, the main parameters required to run the InVEST HQ module include the weights and effective distances of the threat factors as well as the suitability and sensitivity of the habitat for each threat factor. Defining the threat factor parameter for a habitat is a key issue in the InVEST HQ module [32]. Threat factors include biotic factors, abiotic factors, and anthropogenic disturbance. Human activity disturbance has a greater impact on the change of landscape patterning in the study area [25,28]. Considering the intense human activities of cultivated land and construction land, they are regarded as threat factors in this study. According to the User's Guide [25], adjacent degraded non-habitat LULC impose "edge effects" on habitat and can have negative impacts in habitats [25]. Bare land has almost no vegetation coverage, which is not suitable for biological survival, so it is also regarded as a threat factor.

The parameter of threat factors includes weight, maximum distance, and decay type, which were determined by empirical values or expert knowledge. The assignment of weight, maximum impact distance and degradation type of threat factors refer to the existing research results in similar circumstances and the InVEST User's Guide [25,29,33].

As shown in Table 1, human activities are the most intense in urban land, which is more destructive to nearby habitats than other threat factors. Industrial mining and transportation land will destroy vegetation and attract more traffic, so it will also cause greater destruction to nearby habitats. Cultivated land which can be used as habitats for some organisms and rural residential area cause less destruction to nearby habitats. Bare land has the smallest area, which has a limited impact on the nearby habitats and has the potential to develop into a habitat, so its weight is the lowest.

**Table 1.** Attributes of threat data with different land use.

| Threat Factors(Land Use) | Max Distance/km | Weight | Decay |
|---|---|---|---|
| Cultivated land | 5 | 0.6 | linear |
| Urban land | 10 | 1 | exponential |
| Rural residential area | 5 | 0.7 | exponential |
| Industrial mining and transportation land | 6 | 0.9 | exponential |
| Bare land | 4 | 0.2 | linear |

In the InVEST HQ module, the habitat suitability score is a numerical parameter based on LULC types regarded as suitable habitats. Each LULC type gives a habitat suitability score with a value ranging from 0 to 1, where a value of 1 indicates the highest habitat suitability. A ranking of less than 1 indicates habitat where a species or functional group may have lower survivability [25,32]. We assumed each LULC type was suitable according to the habitat suitability score in the InVEST User's Guide [25], with a consideration for overall biodiversity in the study area.

The sensitivity of habitat types to threat factors was a numerical parameter given for each LULC type between each threat factor in the InVEST HQ module, with the value ranging from 0 to 1, with 1 representing high sensitivity to a threat factor. The original parameter of the sensitivity score for LULC to the threat factor within the model were based on the empirical value [32], which generally relies on expert knowledge [28].For the setting of the correlation coefficients between each LULC type and each threat factor, we refer to the existing research results in similar circumstances and InVEST User's Guide [25,28,32].

As shown in Table 2, construction land as non-suitable habitat is set at a value of 0, including urban land, rural residential area and industrial mining and transportation

land. Closed woodland and lakes are treated as (absolute) habitat, which is set as value 1. The land types less affected by human activities, such as bush, high-coverage grassland, swamp, and rivers are used as suitable habitats [25], so they are set high values from 0.8 to 0.9. Other LULC values are based on relevant literature [28,30,32] and InVEST User's Guide [25]. The assignment of relative sensitivity of each landscape to threat factors refer to existing research results and the software user's manual [25,28,32].

**Table 2.** Habitat suitability of different landscape types and sensitivity of LULC types to each threat.

| LULC Type | | Habitat Suitability | Sensitivity to Threat Factors | | | | |
|---|---|---|---|---|---|---|---|
| Primary Type | Secondary Type | | Cultivated Land | Urban Land | Rural Residential Area | Industrial Mining and Transportation Land | Bare Land |
| Cultivated land | paddy field * | 0.25 | 0.3 | 0.5 | 0.4 | 0.5 | 0.4 |
| | Cropland | 0.15 | 0.3 | 0.5 | 0.4 | 0.5 | 0.4 |
| Woodland | Woodland | 1 | 0.8 | 0.9 | 0.8 | 0.8 | 0.5 |
| | Bush | 0.8 | 0.4 | 0.8 | 0.7 | 0.7 | 0.4 |
| | open woodland | 0.6 | 0.85 | 0.9 | 0.8 | 0.8 | 0.5 |
| | other woodland * | 0.4 | 0.9 | 0.9 | 0.8 | 0.8 | 0.5 |
| Grassland | high-coverage grassland | 0.8 | 0.4 | 0.6 | 0.5 | 0.6 | 0.5 |
| Wetland | Rivers * | 0.9 | 0.65 | 0.85 | 0.75 | 0.8 | 0.4 |
| | Lakes * | 1 | 0.7 | 0.9 | 0.8 | 0.7 | 0.4 |
| | Shallows * | 0.8 | 0.7 | 0.9 | 0.8 | 0.7 | 0.4 |
| | beach | 0.6 | 0.75 | 0.95 | 0.85 | 0.7 | 0.4 |
| | beach land | 0.6 | 0.75 | 0.95 | 0.85 | 0.7 | 0.4 |
| | Swamp | 0.9 | 0.7 | 0.8 | 0.75 | 0.6 | 0.5 |
| Construction land | urban land * | 0 | 0 | 0 | 0 | 0 | 0 |
| | rural residential area * | 0 | 0 | 0 | 0 | 0 | 0 |
| | Industrial mining and transportation land * | 0 | 0 | 0 | 0 | 0 | 0 |
| Bare land | bare land | 0.1 | 0.1 | 0.3 | 0.3 | 0.3 | 0.2 |
| | barren land | 0.1 | 0.1 | 0.3 | 0.3 | 0.3 | 0.2 |

Note(s): * is land use status in the Grand Canal (Suzhou section).

## 4. Results

Land use data (2010 and 2018) analyzed by ArcGIS show eight types of land use, of which urban land is the main type of land for land use in the Suzhou section of the Beijing-Hangzhou Grand Canal. A series of habitat quality maps of study area were carried out using the InVEST-HQ model in different years, with the habitat quality scoring between 0 and 1 in each cell (higher values indicating a more suitable habitat). Habitat quality is spatially low in the middle and high on both sides as shown in Figure 6. Habitat degradation is lowest in the middle and highest downstream (more vulnerable to ecological risks), as shown in Figure 7.

### 4.1. Land-Use Change between 2010 and 2018

From 2010 to 2018, the rapid expansion of urban land and the continuous reduction of rural settlements were the main features of land use change in the Suzhou section of the Grand Canal. The land use of the Suzhou section of the Grand Canal was mainly urban land, and the large area of land use type changed from cultivated lands to urban construction lands in 2010 and 2018. The area of urban land increased by 6% (320.76 km$^2$), from 2010 to 2018, whereas the paddy fields decreased by 19% (302.67 km$^2$). To a large extent, the rapid expansion of urbanization in Suzhou has caused the total amount of ecological resources and the natural environment to be squeezed [33].

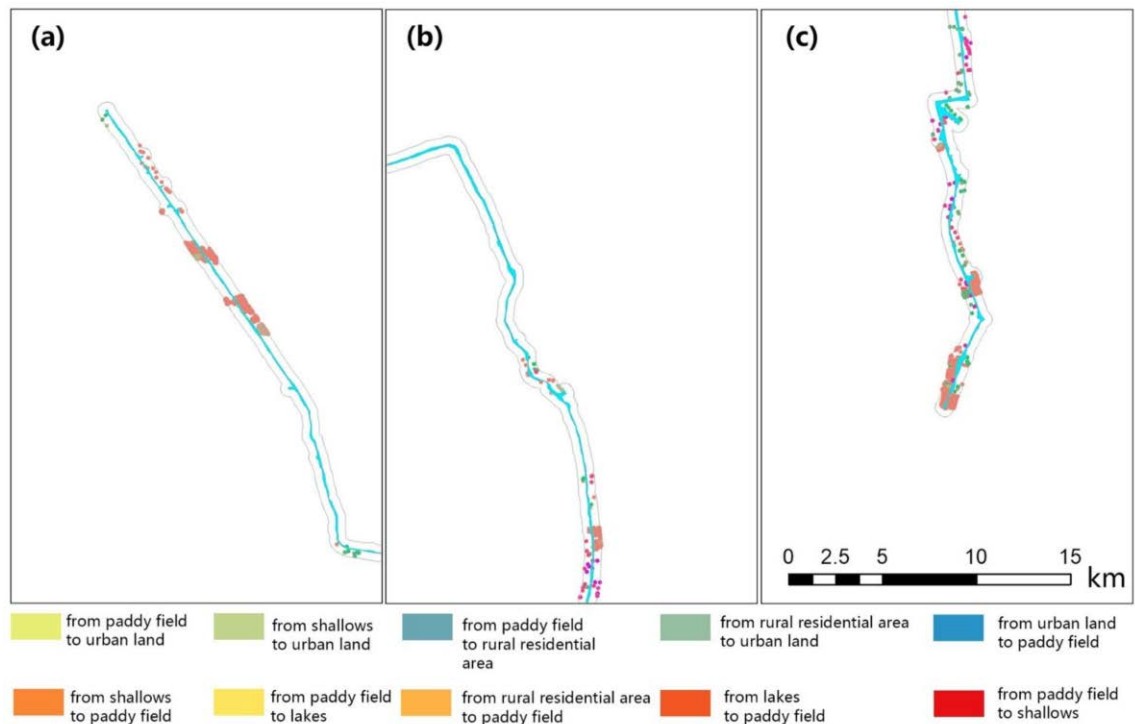

**Figure 6.** Changes of land use in the Suzhou section of the Grand Canal between 2010 and 2018. (**a**): Land type change in the upper reaches of the Grand Canal (Suzhou section), (**b**): Land type change in the middle reaches of the Grand Canal (Suzhou section), (**c**): Land type change in the downstream reaches of the Grand Canal (Suzhou section).

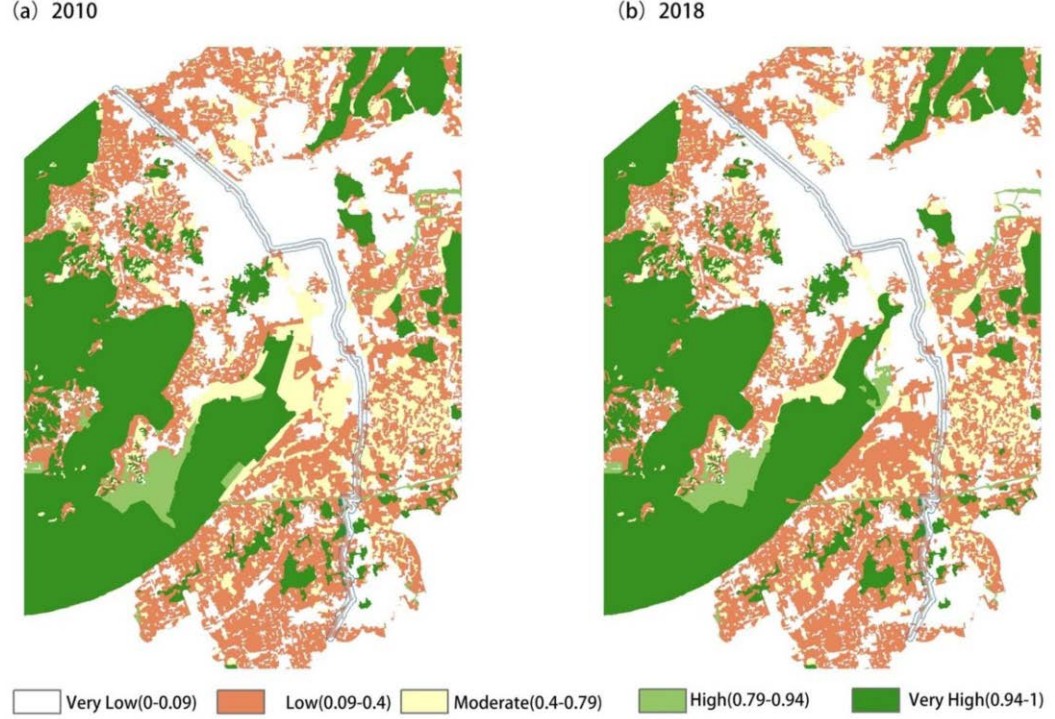

**Figure 7.** Habitat quality map of the study area between 2010 and 2018 (the middle section is mainly low-quality habitats, while the moderate and high-quality habitats are distributed in the upstream and downstream of the Suzhou section of the Grand Canal).

*4.2. Changes in Habitat Quality between 2010 and 2018*

Based on the distribution of land use in Suzhou in 2010 and 2018, the Habitat Quality module analysis of the InVEST 3.9.0 model shows the changes in habitat quality, which is low in the middle and high on both sides. Through the natural breaks in ArcGIS, classification can be divided into the following five categories of Habitat Quality: Very Low (0–0.09), Low (0.09–0.4), Moderate (0.4–0.79), High (0.79–0.94), and Very High (0.94–1). As shown in Table 3, there is a trend of differentiation: there is an increasing change in the very low degree of habitat quality (urban land) and the very high degree of habitat quality (paddy fields, rivers, and lakes), and the other levels of habitat quality show a downward trend. Most of the habitat quality of the Suzhou section of the Grand Canal is in the moderate and low grades and shows a downward trend as the area of moderate habitat quality decreased by 7% and the area of low habitat quality decreased by 18.9%.

**Table 3.** Area and proportion of different levels of habitat quality.

| Different Habitat Qualities | Area/km$^2$ | | Change | Range of Change |
|---|---|---|---|---|
| | **2010** | **2018** | | |
| Very Low Habitat Quality (0–0.09) | 5442.3 | 5762.88 | 320.58 | 5% |
| Low Habitat Quality (0.09–0.4) | 1597.77 | 1294.56 | −303.21 | −18.9% |
| Moderate Habitat Quality (0.4–0.79) | 248.4 | 230.49 | −17.91 | −7% |
| High Habitat Quality (0.79–0.94) | 53.82 | 52.74 | −1.08 | −2% |
| Very High Habitat Quality (0.94–1) | 321.57 | 323.19 | 1.62 | 0.5% |

*4.3. Changes in Habitat Degradation Index between 2010 and 2018*

Based on the Habitat Quality module of InVEST 3.9.0 to assess the change status of habitat quality, the habitat degradation index map can also be produced as shown in Figure 8. The assessment results are divided into three categories according to the Equal Interval method in ArcGis, as shown in Figure 9: low degradation index zone (0–0.02), medium degradation index zone (0.02–0.08), and high degradation index zone (0.08–0.19).

The decline in habitat quality is thought to be the result of increased land use intensity. The spatial variation of habitat degradation reflects the severity of regional habitat quality degradation. From the perspective of spatial distribution, the middle and upper reaches of the Grand Canal had a higher degree of urbanization, showing a trend of low habitat degradation. Additionally, the areas with the highest habitat degradation index are in the downstream of the Grand Canal, which experiences a greater ecological risk.

It can be seen from a certain type of land (except residential areas) with a large enough area that the central area has a low degree of degradation, and the edge area has a high degree of degradation.

The habitat with a low degradation dominates the Grand Canal's middle section, whereas the habitat with a moderate degradation dominates the upper and lower sections. Most of the areas with the high degree of degradation are in the downstream area of the Suzhou section of the Grand Canal.

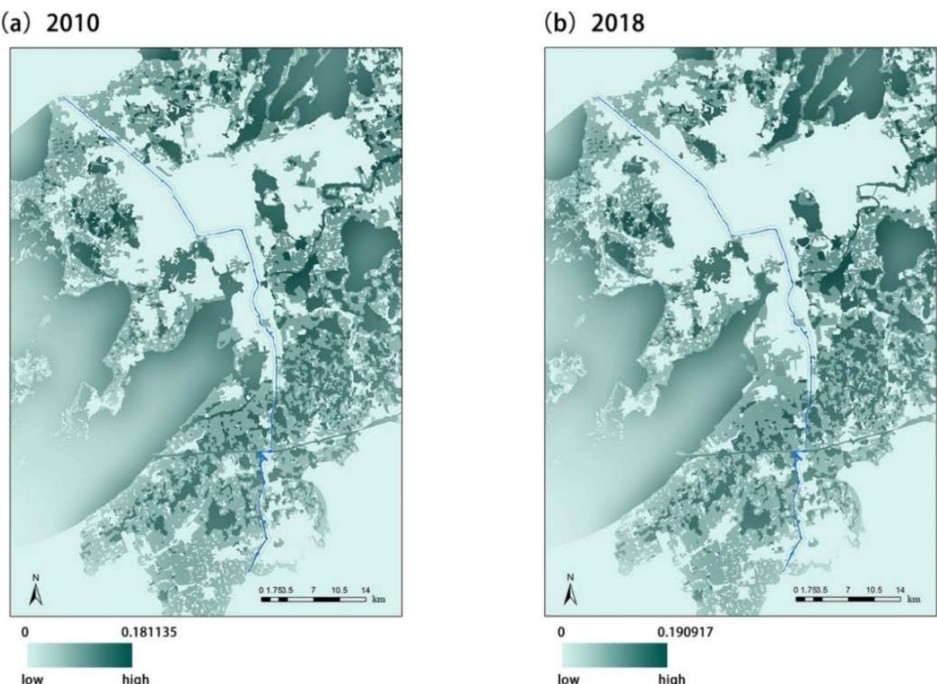

**Figure 8.** Index map of habitat degradation in study area between 2010 and 2018.

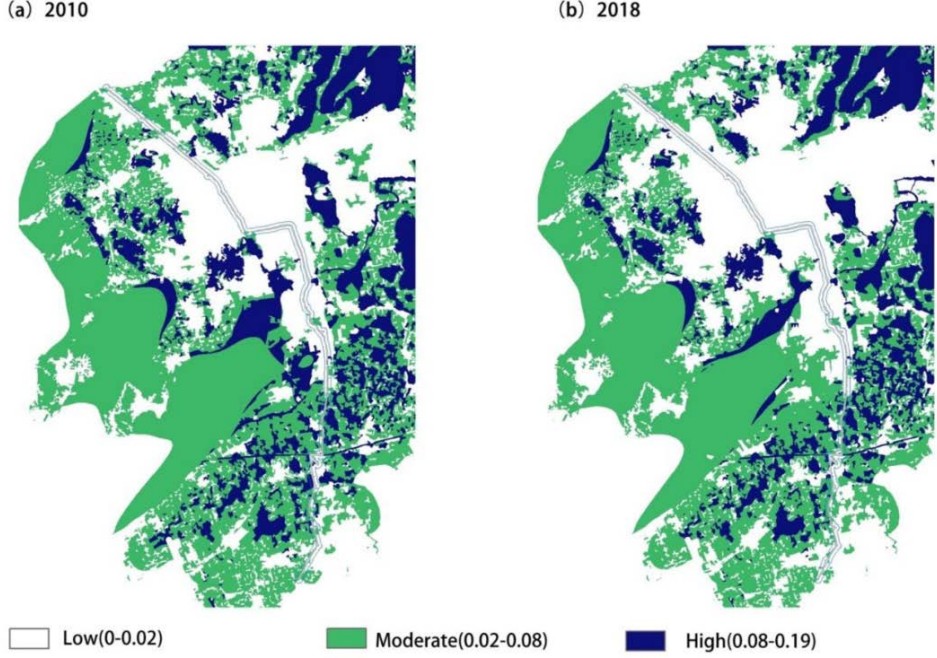

**Figure 9.** Equal interval habitat degradation grading map of study area between 2010 and 2018.

*4.4. Ecological Risk Prediction*

Urban ecological risk is particularly complicated since it is impacted by human activities, complex exposure routes, and various sources, receptors, and stakeholders [34]. From the perspective of environmental receptors, this study analyzes the sensitivity of the environment to ecological risks through its habitat quality and habitat degradation degree. The average ecological risk caused by newly developed land is generally higher than the average ecological risk caused by the total amount of developed land [35]. Additionally, newly developed land usually occurs in low-development areas (farmland, paddy fields), which are highly degraded areas. Therefore, the first we need to guard against

is not areas with low habitat quality such as urban land, but areas with a high degree of degradation (farmland, paddy fields). Such areas are mainly distributed in the downstream of the Suzhou section of the Grand Canal. The next warning is the upper reaches of the Grand Canal, followed by the middle reaches. To address ecological risks, we propose combined strategies such as urban rewilding, ecological revetment, recovering vegetation for increasing plant richness, and improving biodiversity in next section of the Grand Canal.

## 5. Discussion

### 5.1. Potential Sources of Risks and Early-Warning Strategies

From 2010 to 2018, the habitat quality index of the Grand Canal (Suzhou section) was generally in good condition, but the polarization between upstream and downstream was obvious. The area of high-quality habitat in the downstream of the river showed a decreasing trend, which is due to the increased development of Wujiang district (which is in the downstream of the Grand Canal) [36]. The areas with the lowest habitat quality are located in the middle of the Grand Canal (mainly towns). The distribution and changes in habitat quality indicate that the faster the economic development, the lower the habitat quality. Priority should therefore be given to protecting the downstream area of the Grand Canal by coordinating urban development and ecological protection [36].

From 2010 to 2018, the habitat degradation index of the Suzhou section of the Grand Canal showed an obvious decline trend in the downstream of the Grand Canal. Since the downstream had a decreased natural water area, an increased wetland area, and a decreased coverage of the grassland and forest from 2010 to 2018, the land use change might contribute to the decline of habitat quality and the threat of biodiversity. Due to the high habitat degradation index, the downstream of the Grand Canal must be prioritized in the face of ecological threats. The distributions and changes of habitat degradation show that the faster the economic development, the more vulnerable the habitat quality is to damage.

Habitats with low degradation are typically found in the middle of lands. The further out, the higher the degree of habitat degradation (especially at the edge of the lands, basically showing a high degradation state). High degradation areas are generally located at the junction of different lands and are more susceptible to the impact of other lands. The areas with high habitat quality generally have lower degradation index zones, except residential areas [33,36]. The interesting discovery is that the smart urban expansion scenario might mitigate or even eliminate the negative effects of urbanization, as built-up areas show low degradation indicating that a compromise between habitat protection and urban growth is possible [29,33,36].

Under the influence of climate change, land use change and human activities, the state of an ecosystem will change from one steady state to another [21]. Due to the complexity, nonlinearity, randomness and other characteristics of environmental pressure, the state transition is often characterized by nonlinearity, sudden change, and jump change. Defining the inflection point or threshold point of the system state transition and capturing the changes in ecosystem structure and properties before the critical inflection point is an early warning [37,38].

This study adopts a series of ecological restoration methods to improve the ecological resilience of the Suzhou section of the Grand Canal, to increase the response time of the ecosystem to ecological risks, namely early-warning strategies.

### 5.2. Proposed Restoration Strategies for Different Habitat Degradation Index Areas

Three types of degraded areas have different responses to ecological risks, as shown in Figure 10. Previous research [39,40] applied ecological zoning to adopt corresponding protection methods for different zones. According to the three types of habitat degradation, a hierarchical protection strategy was adopted to effectively reduce ecological risks. Although the proposed strategies cannot completely reverse the negative impact of urban-

ization at a large scale, they can play an ecological role for the mitigation of anthropogenic disturbances at a local scale.

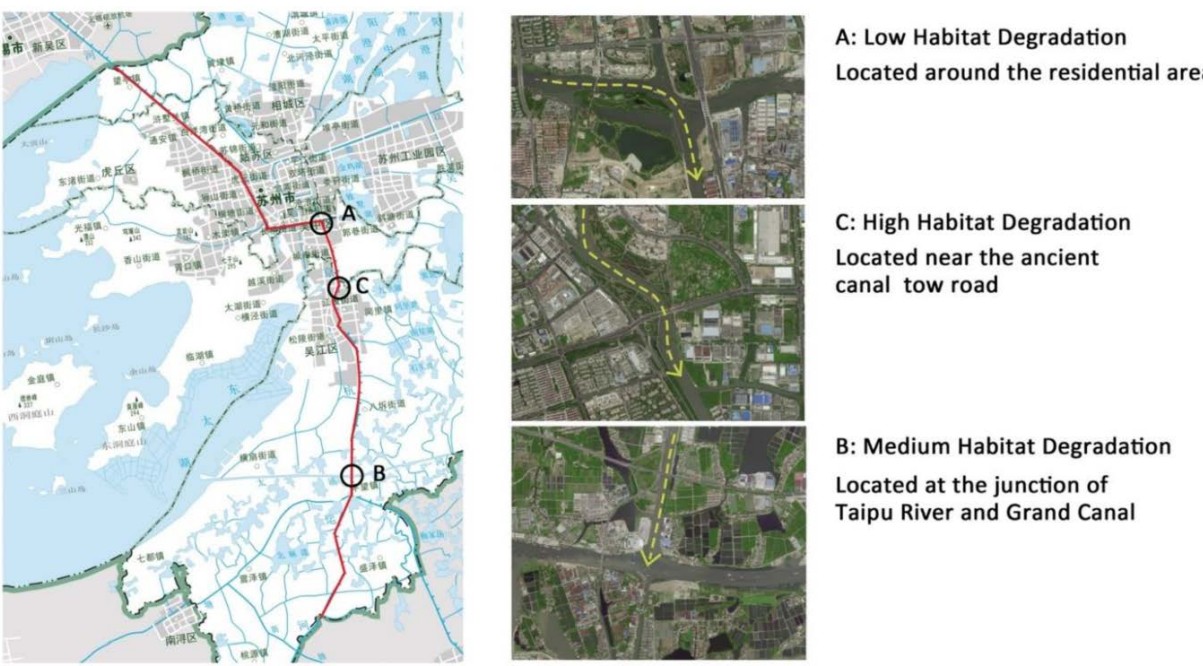

**Figure 10.** Three types of habitat degradation area within the Suzhou section of the Grand Canal.

When subjected to sudden disturbances (such as flood and the leakage of flammable), compared with low-degraded and moderate-degraded areas, high-degraded habitats are more easily affected and have weaker autonomous recovery capabilities. Therefore, artificial restoration is the main method to strengthen the ability of regional ecosystems to recover and to quickly restore the regional ecosystem balance.

When subjected to cumulative disturbances (such as heavy metal pollution), low-degraded areas autonomously maintain ecosystem balance using their recovery capabilities. However, high-degraded areas do not have a good recovery ability (they will take longer time to be restored), so manual intervention is required to restore the balance of the ecosystem through artificial restoration [29].

Basically, habitats with a low degree of degradation are more likely to be recovered from ecological threats and to regain stability for both circumstances. Therefore, different combined modes of natural restoration and artificial restoration can be adopted to recover different degraded areas, as shown in Table 4. Since the areas with low degree of degradation are easy to be restored, we choose the strategy of natural restoration combined with the method of nature-based-solutions (NBS) and urban rewilding. Nature-based solutions use nature's strength to transform environmental, social, and economic concerns into opportunities for innovation [41]; for example, the revetment can be fixed by the roots of the plant. The rewilding in urban informal greenspace can help to mitigate climate change and biodiversity loss [42], while increasing ecological system's resilience to risks and threats. Habitats with high degradation are vulnerable to ecological risks and are difficult to restore on their own. Therefore, manual intervention with artificial repair (such as artificially constructed ecological revetments) is required. For areas with moderate degradation, a combination of methods for natural restoration and artificial restoration could be applied. For instance, urban green spaces (UGS) and vegetation have the potential to aid in the restoration of ecosystem services [43] and biodiversity in cities [44]. The aim is to create more dynamically self-regulating landscapes [45].

**Table 4.** Ecological restoration methods for different degraded areas.

| Degree of Degradation | Strategy | Method |
|---|---|---|
| Low degradation index zone | natural restoration | NBS: Rewilding; |
| Moderate degradation index zone | Natural Repair + Artificial Repair | NBS: Reforestation; Improve biodiversity + Ecological revetment |
| High degradation index zone | Artificial Repair | Increase plant richness; Improve biodiversity |

5.2.1. Proposed Restoration Strategies for Low Habitat Degradation Area A

The low-degraded areas have a better ability to autonomously recover ecosystems, so they can recover well from threats subjected to intermittent disturbances and transient disturbances. The low habitat degradation zone we chose is located around the residential area with high level of urbanization. Although the degree of degradation is low in such regions (due to their location in the center of just a land type), the quality of the habitat is also low. As a result, urban rewilding could be adopted increase habitat quality while maintaining a low degradation index. Rewilding is a call for boosting ecological complexity in conservation [42,46].

5.2.2. Proposed Restoration Strategies for Moderate Habitat Degradation Area B

The moderate-degraded habitats perform well in the face of environmental threats (sudden and cumulative disturbances). The medium habitat degradation site we choose is located at the junction of the Taipu River and the Grand Canal. As this area is densely covered with many small lakes and rivers, we intend to carry out ecological restoration on the lake riparian zones. Through transforming the hard revetments to ecological revetments and planting with a variety of waterfront plants in the buffer zone [44], it is expected that the biodiversity will be enhanced [47,48] and urban heat island effect can be reduced [49].

5.2.3. Proposed Restoration Strategies for High Habitat Degradation Area C

The high-degraded areas are vulnerable to changes caused by changes in the external environment due to their sensitivity. It is more difficult to return to a stable state towards ecological threats. The high habitat degradation zone we choose is near the ancient canal tow road. The habitat quality in this region is medium, and the high degradation habitats are located at the junction of green spaces and residential areas. Previous studies underline the importance of conserving green space for biodiversity conservation. It is believed that we can increase the green spaces on the street corner and plant plants to increase the abundance of plants and thus the stability of the ecosystem [45].

**6. Conclusions**

In recent years, with the rapid advancement of industrialization and urbanization, the emission of pollutants in the areas along the Beijing-Hangzhou Grand Canal has increased, and the problem of non-point source pollution from shipping and agriculture has become prominent. At the same time, some prominent and complex environmental problems, such as the inability of water quality in some river sections to consistently meet the standard, have become increasingly prominent. Therefore, this study takes the Suzhou section of the Grand Canal as a research sample to explore its habitat quality, ecological risks, and early-warning strategies, to achieve the purpose of strengthening the ecological protection and restoration of the Grand Canal. In addition, the research methods and research results can be further applied to research in river basins in other regions of China and even other countries.

*6.1. Habitat Quality and Habitat Degradation Index*

Based on the invest model, this study evaluated the habitat quality and habitat degradation in the 400 m buffer zone of the Suzhou section of the Grand Canal in 2010 and 2018 and their changing trends. The conclusions are as follows: (1) Between years 2010 and 2018, the habitat quality of the Suzhou section of the Grand Canal was at the levels of the

moderate and lower grades and showed a trend of continuous decline. Habitat quality showed a trend of continuous increase in low-quality areas and continuous decrease in high-quality areas. From the perspective of the spatial pattern of habitat quality, it has the overall characteristics of low in the center and high in the north and south. (2) From 2010 to 2018, the degradation degree of the Suzhou section of the Grand Canal showed low in the middle and high at both ends in terms of spatial pattern. The areas with low degradation were generally located in the middle areas. The closer to the center of a type of habitat, the lower the degree of degradation. (3) According to the level of habitat quality and the degree of habitat degradation, corresponding ecological protection strategies are put forward.

*6.2. Ecological Risk Prediction and Early-Warning Strategies: Improvement of Ecosystem Stability*

The habitat quality of the upper and downstream of the Suzhou section of the Grand Canal is significantly polarized. In recent years, the natural water area in the downstream of the river has decreased, wetlands have increased, grassland and woodland cover have decreased, habitat quality has declined, and biodiversity has been threatened. The decline in habitat quality will increase ecological risk while reducing ecological resilience. After investigation and analysis, the main ecological risks faced by the Suzhou section of the Grand Canal are a series of risks brought about by water pollution and biodiversity reduction. Therefore, according to the specific environmental characteristics of the Suzhou section of the Grand Canal, we propose a series of strategies to improve its ability to cope with ecological risks.

To address ecological risks, strategies including urban rewilding, ecological revetment, recovering vegetation for increasing plant richness, improving biodiversity are proposed to improve habitat quality and reduce habitat degradation and ecological vulnerability [50].The ecologically friendly artificial texture can be used along nature-based solution approach for ecological restoration, with the combination of recovering community structure and establishing ecosystem functioning [51–53].These measures help to improve habitat quality and reduce habitat degradation. Practically: (1) urban rewilding is recommended in the low-degraded areas to increase habitat quality and maintain a low degradation index [42,46]; (2) in the medium-degraded areas, ecological revetment and vegetation can be chosen to increase biodiversity by creating habitats for wildlife, thereby achieving the purpose of improving ecological resilience [44]; and (3) in the high-degraded areas, increase greenness on the street corner and plant vegetation to increase the abundance of plants. These measures will improve the stability of the ecosystem by improving habitat quality and reducing habitat degradation index. Ecological risk prediction and risk early-warning are often uncontrollable to a certain extent [54], so we must strengthen the ability to reduce ecosystem vulnerability and to enhance ecosystem resilience to cope with multiple stressors induced by ecological risks.

**Author Contributions:** Y.Z.: conceptualization, methodology, funding acquisition, supervision, writing and editing; C.Z., X.Z., X.W. (Xinge Wang), T.L., Z.L., Z.J., X.W. (Xinyue Wang), Q.H. and W.S.: methodology, data curation, investigation, analysis, writing original draft; Q.L., L.T., J.W., G.Z., Y.T., J.Z., J.H., Y.S. and F.M.: review. All authors have read and agreed to the published version of the manuscript.

**Funding:** National Key Research and Development Program of China (Grant No. 2021YFE0200100); Suzhou Grand Canal Cultural Zone Construction Research Institute (Grant No. 21SZDYH103).

**Institutional Review Board Statement:** Not applicable.

**Informed Consent Statement:** Not applicable.

**Acknowledgments:** We would like to thank reviewers for providing constructive comments and suggestions. We are very grateful to Xuerong Ma for their technical help. This work is supported by the National Key Research and Development Program of China (Grant No. 2021YFE0200100,

China-Portugal Belt and Road Cooperation Laboratory of Cultural Heritage Conservation Science), and Suzhou Grand Canal Cultural Zone Construction Research Institute (Grant No. 21SZDYH103).

**Conflicts of Interest:** The authors declare no conflict of interest.

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
