# Peer review of "Habitat Quality Assessment and Ecological Risks Prediction: An Analysis in the Beijing-Hangzhou Grand Canal (Suzhou Section)"

_water, doi:10.3390/w14172602_

Round 1
Reviewer 1 Report
The paper is interesting and deals with current problems related to the protection of environmental resources. The methods were applied appropriately to the topic discussed. The test results were properly described.
I only have one comment. As the discussed topic is of high importance for the modern world, it would be good to emphasize the universality of the obtained results in the conclusions.
Reviewer 2 Report
The English of the manuscript needs to be considerably revised, before the scientific content of the manuscript can be judged.
The English of the manuscript is not of adequate quality for the journal water. I would strongly suggest that the manuscript is proofread and corrected by a professional proof-reading service to provide a manuscript in grammatically correct and well understandable English and in order to not waste the time of the reviewers. English correction is not the task of the reviewers and the authors have to take care of providing a manuscript in well understandable and grammatically correct English before submission to a journal. I suggest to reject the manuscript with possibility of resubmission.
Reviewer 3 Report
The main problem with the paper by Yixin Zhang et al. is that it is hardly understandable for people (like myself), who did not use and are not familiar with the InVEST software. Data presented in Tables 2 and 3 within Material and methods section are referred by Authors to “existing research results and to software user manual”. It is not clear, however, what do they mean, how they were acquired and, if based on results, why are they presented in Material and methods section. There is column “Habitat suitability” in Table 3 but no information for what this habitat is suitable. Title of the table announces presentation of “relative sensitivity of each land type to threat factors” but head of the columns is “threat factors”. As a result, potential reader obtains a large set of parameters of unknown origin and importance. In view of the above doubts, I think the habitat quality was not assessed (as given in the title) but rather determined based on land use type. Therefore, the title is somehow misleading.
The same section shows the study area (Fig. 3) with geographic coordinates which indicate that the Suzhou Municipal District is mostly situated outside this area (line 156). BTW – does a coordinate 30o86’ N mean 31o26’?
Presentation of results is far from being perfect. From Fig. 4 one can easily distinguish various land use types kilometers apart from the Canal but doing it for the study area (400 m buffer zone) is impossible. The latter pertains also to Fig 5, whose legend presents 10 types of land use changes but only two or three can be seen in graphs. Results described in lines 278-279 disagree with those presented in Table 4.
In the Discussion and Conclusions Authors propose some restoration strategies for various types of habitat degradation. I am afraid that none of the proposed actions like urban rewilding by the creation of “green spaces on the street corner” (line 426) or planting shorelines of water bodies (lines 416-417) will prevent from damages caused by expanding road networks and residential/industrial built-up areas.
Specific remarks
- In lines 128-136 Authors properly defined possible risks any ecosystem may experience. Why these risks were not referred to in Discussion and Conclusion sections?
- It would be nice to have acronyms given in full when first mentioned (DEM in line 177 or LULC in line 221).
- I would suggest using “very low” and “very high” instead of “lower” and “higher” in the legend of Fig. 6.
- What is the meaning of “Shallows” and “Other woodland” in the legend of Fig. 4?
Language
Presented text needs thorough checking, preferably by an English native speaker (see for example lines 52, 67, the whole caption to Fig. 1, lines 176-177, line 222, 225, 362 , 421).
Round 2
Reviewer 2 Report
The authors have completely ignored my request to improve the English and have the manuscript proof read and corrected by a professional proof-reading service. This is unacceptable. I am not willing to perform an English correction and the manuscript has to be submitted in grammatically correct and well understandable English before submission. No reputable journal that I am aware of allows for English correction after acceptance. I am willing to provide a detailed review of the manuscript when I receive a well-written version without major grammatical errors.
Furthermore, the authors did not provide a point by point response to the authors criticisms. This is necessary to do so in and to do it in English.
Author Response
The manuscript has been edited by a native-English speaker (professional proof-reading service) during this revision process.
Reviewer 3 Report
The second version of the manuscript has been corrected and much improved compared with the first version. I would only suggest correcting minor printing/linguistic errors in lines 40, 52, 67 (physical and chemical not physio-chemical), 81 (what does it mean "flammable") and 497 (please, insert "to).
Author Response
In this revision, we revised the manuscript by following the reviewer's comments.
Reviewer 3: "The second version of the manuscript has been corrected and much improved compared with the first version. I would only suggest correcting minor printing/linguistic errors in lines 40, 52, 67 (physical and chemical not physio-chemical), 81 (what does it mean "flammable") and 497 (please, insert "to). "
Answer: We changed these in the new revision.